# A mixed-methods study of factors influencing postpartum intrauterine device uptake after family planning counseling among women in Kigali, Rwanda

**Mariama S. Tounkara**[1], **Rosine Ingabire**[2], **Dawn L. Comeau**[1], **Etienne Karita**[2], **Susan Allen**[3], **Julien Nyombayire**[2], **Rachel Parker**[3], **Lisa B. Haddad**[4], **Vanessa Da Costa**[5], **Amanda Tichacek**[3], **Amelia Mazzei**[2], **Jeannine Mukamuyango**[2], **Kristin M. Wall**[5]*

1 Department of Behavioral, Social and Health Education Sciences, Rollins School of Public Health, Emory University, Atlanta, Georgia, United States of America, 2 Projet San Francisco, Department of Pathology & Laboratory Medicine, School of Medicine, Emory University, Kigali, Rwanda, 3 Department of Pathology & Laboratory Medicine, School of Medicine, Emory University, Atlanta, Georgia, United States of America, 4 Center for Biomedical Research, Population Council, New York, New York, United States of America, 5 Department of Epidemiology, Rollins School of Public Health, Emory University, Atlanta, Georgia, United States of America

* kmwall@emoy.edu

**Data Availability Statement:** All relevant data are available on Harvard Dataverse: https://doi.org/10.7910/DVN/GMXIMW.

## Abstract

### Introduction

Rwanda has high unmet need for family planning (FP), especially in the postpartum period when women are advised to space pregnancies at least two years for improved maternal-child health. Despite interest in the copper intrauterine device (IUD), a highly cost-effective method, access and uptake remain low. This study aimed to determine factors associated with postpartum IUD (PPIUD) uptake after postpartum family planning (PPFP) counseling as well as provider perceptions of facilitators and barriers to clients' PPIUD uptake.

### Methods

Postpartum women who received PPFP counseling and were less than 6 weeks postpartum were recruited for a case-control study in Kigali, Rwanda in 2018. We recruited n = 74 women who had accepted and n = 91 women who had declined the PPIUD. Multivariate logistic regression analyses evaluated associations between women's socio-demographics, FP knowledge and decision-making, and the outcome of PPIUD uptake. Six focus groups (FGs) were conducted with FP providers (n = 24) and community health workers (n = 17) trained to deliver PPFP counseling to assess perceptions of PPFP counseling and facilitators and barriers to PPIUD uptake. FG discussions were recorded, translated, and analyzed for themes.

### Results

Factors associated (P<0.1) with PPIUD uptake included citing its non-hormonal nature, effectiveness, and duration of protection against pregnancy as advantages. Exclusive male

**Funding:** This work was supported by the Bill & Melinda Gates Foundation [OPP1160661]. Additional support came from the Emory University Research Council Grant [URCGA16872456], Emory Global Field Experience Award, the Emory Center for AIDS Research [P30 AI050409], the National Institutes of Health [NIAID R01 AI51231; NIAID R01 AI64060; NIAID R37 AI51231], and Emory AITRP Fogarty [5D43TW001042]. The funders had no role in study design, data collection and analysis, decision to publish, or preparation of the manuscript.

**Competing interests:** The authors have declared that no competing interests exist.

partner control over FP decisions (relative to women's control or joint decision-making) was associated with non-use. Overall, limited knowledge about some aspects of the PPIUD persisted among clients even after counseling. Provider FGs highlighted client concerns, inconsistent FP messaging, and lack of male partner involvement as factors influencing non-use.

## Conclusions

Knowledge of the IUD and its benefits was associated with PPIUD uptake. There is need to refine PPFP counseling messages to address remaining knowledge gaps and concerns. Additionally, male partner involvement in FP counseling and decisions with their partners could be a key strategy to increase both PPIUD and FP uptake in Rwanda.

## Introduction

Although increased family planning access has reduced global fertility rates, unintended pregnancies, and unsafe abortions, these outcomes are still relatively frequent in sub-Saharan African countries where unmet need for family planning is common [1]. Unmet need for postpartum family planning (PPFP) is particularly high throughout sub-Saharan Africa [2].

The World Health Organization (WHO) recommends a birth spacing interval of two years for improved maternal-child health [3]. In Rwanda, 52% of pregnancies occur less than two years after the preceding birth [4], 47% of all pregnancies are unintended [5], and 51% of postpartum women have an unmet need for family planning [4]. To ameliorate these issues, Rwanda is spearheading PPFP programs with a particular focus on the long-acting reversible contraception (LARC) methods [6, 7]. LARCs (non-hormonal copper IUDs and hormonal implants) are highly effective, convenient, safe, easy to use, and require infrequent follow-up visits relative to the shorter acting methods [8–10]. Despite IUDs being less expensive and more cost-effective than other contraceptives, knowledge [11] and uptake are still extremely low and this warrants further exploration [12].

Community health workers (CHWs) and providers can offer counseling to educate women about PPFP, including LARCs 1) during routine antenatal care (ANC) visits, 2) during or after labor and delivery (L&D), 3) pre-discharge/post-partum and 4) at infant vaccination visits [13]. LARCs can be provided while women are at the health facility during delivery or at infant vaccination visits. Convenient times to insert the copper IUD while women are at the health facility include post-placental (within 10 minutes of placental delivery), postpartum (within 48 hours of delivery), at 4–6 weeks postpartum, and during infant vaccination visits [14–16]. However, of the 30% of women using modern contraceptive methods in Rwanda, only 3% use the copper IUD while 17% use the implant [7], which is more well-known relative to the IUD. Among post-partum Rwandan women who are using a family planning method, only 2% use an IUD [4].

Projet San Francisco (PSF) developed a multi-level intervention to improve PPFP services in Rwanda with a focus on PPLARC which was implemented in August of 2017 [13]. Based on input from stakeholders, providers, CHWs, and couples/clients, a promotional and educational flipchart was developed to educate women about the PPIUD (along with the full menu of PPFP contraceptive options) to be delivered to women or couples during routine ANC, labor, within 48 hours postpartum, infant vaccination services and in the community by CHWs [13]. Providers from 6 government facilities including hospitals and clinics went through a 2-day didactic session followed by practical training including PPIUD insertion and

removal, mock counseling sessions, and post-training tests. CHWs who worked with pregnant women and newborns received 1-day training to review the flipchart. After the first year of implementation, there was a PPIUD acceptance rate of 29% among 9073 women who received one-on-one PPFP counseling [13].

The objectives of the present study were to evaluate factors associated with PPIUD uptake after receipt of PPFP counseling and to assess the barriers and facilitators to PPIUD promotion and uptake as perceived by the PPFP providers and CHWs.

## Methods

In this mixed-methods study, quantitative data from a case-control study and qualitative data from focus group discussions (FGDs) were collected and analyzed.

### Sample and setting

Convenience samples of participants for the case-control study (with a target recruitment ratio of 1:1) and FGDs were recruited at three government health facilities in Kigali, Rwanda in 2018. Sites were selected because they are high-volume facilities where our PPFP counseling and provision [13] had been implemented almost a year prior [17]. PPFP counseling included information on health benefits of spacing pregnancies, facts (including benefits and side-effects) about all PPFP methods including the IUD, and description of the PPLARC insertion procedures.

Eligible participants for the case-control study must: 1) have received PPFP promotions and delivered at any of the PSF-affiliated facilities; 2) speak Kinyarwanda (the local language); and 3) have voluntarily agreed to participate and provided written informed consent. Verification of the women having previously received our counseling intervention was ascertained by checking FP promotion and insertion government logbooks at each facility.

Eligible participants for the FGDs were PPFP trained and certified providers (physicians, nurses, midwives) and CHWs who had been promoting PPFP use for at least 4 months prior to FGDs.

### Recruitment

For the case-control study PPIUD users were recruited face-to-face during PPIUD follow-up visits and 6-week infant vaccination services at the clinic. Nonusers (women who used a non-LARC method or no method at all) were selected from postpartum and infant vaccinations visits. Providers for the FGDs were recruited during work breaks and after work. CHWs were recruited during their monthly check-in meetings with nurse counselors.

### Data collection

**Quantitative strand.** Two separate semi-structured surveys (one for PPIUD users and another for nonusers) included closed-ended questions to assess sociodemographic characteristics, contraceptive use history, reproductive history and sources of contraceptive information. Two questions assessed knowledge of timing related to PPIUD insertion after counseling–one asked for a non-prompted, spontaneous response from participants while the other provided women with a list of possible responses that they could choose from. Open-ended questions assessed knowledge about postpartum family planning, PPIUD use, benefits, and facts as well as family planning decision-making factors. PPIUD users were also asked about expulsions and side-effects. On a five-point Likert-scale, PPIUD users were asked about

the service delivery environment and overall satisfaction with the intervention and the method. PPIUD nonusers were asked about future contraceptive plans.

The surveys were revised with the help of PSF staff and translated from English into Kinyarwanda by native speakers to ensure content and semantic equivalence. They were pilot tested among six PPIUD users and eight nonusers and then iteratively revised to improve question phrasing, order, and for overall linguistic comprehension and cultural propriety. The surveys were administered during regular government health facility hours by four trained PSF counselors using tablets through the digital survey platform, Survey CTO (Dobility, Cambridge, USA). Surveys took 10–15 minutes to deliver.

**Qualitative strand.** The FGD guides were developed from the PPFP counseling tools and observations made during promotions and based in grounded theory. The questions explored the providers' perceptions of the facilitators and barriers to the intervention and client PPIUD uptake. All six FGDs (two consisting of 17 CHWs and four of 24 providers) were led by trained PSF staff and nurse counselors. FGDs were conducted from July to August 2018 and ranged from 30 minutes to 2 hours in duration. FGDs were audiotaped and detailed notes were taken to identify themes.

## Quantitative data analysis

Data was checked for completeness, cleaned, and coded for analysis. Quantitative analyses were performed using SAS version 9.4 (SAS Institute, Cary, NC, USA). Age was estimated by subtracting the year of birth from the year of data collection (2018).

Qualitative responses to the open-ended questions were quantized [18]. This conversion of qualitative into quantitative data allowed for statistical assimilation with the other obtained quantitative data. Data regarding participant and partner control in FP decision making was collected with three response categories: woman alone, man alone, or joint control. The point estimates for woman alone and joint control were both significant and of similar magnitude and direction relative to the null; thus, we decided to collapse those two categories.

Numerical and categorical responses were reported as means and standard deviations and frequencies and percentages, respectively. These data were stratified by the outcome of interest which was current PPIUD use versus nonuse. This was followed by bivariate analyses to test associations between sociodemographic, reproductive characteristics, PPIUD knowledge, decision-making factors, and the outcome of PPIUD uptake. Chi-square tests (or Fisher's exact tests) and independent t-tests were used to assess for whether covariate distributions were statistically significantly different by the outcome. Variables that significantly differed by outcome status ($P<0.1$) in bivariate analyses were evaluated in crude logistic regression models and prevalence odds ratios (ORs), 95% confidence intervals (CIs), and p-values were obtained. The variables were checked for evidence of multicollinearity based on a Spearman correlation coefficient of $>0.8$ to ensure that the predictors were not highly associated with each other. Selection for the adjusted multivariate logistic regression model was considered at $P<0.1$ due to the formative nature and relatively limited sample size of this study. Backwards selection methods in SAS were used to obtain the final logistic regression model [19]. Variables attaining significance at $P<0.1$ in the multivariate analysis were retained for the final adjusted logistic model.

## Qualitative data analysis

Thematic analysis was used to identify the perceptions of PPIUD uptake in the FGDs by two analysts. Themes derived from the data related to facilitators and barriers to uptake that contextualized the quantitative survey findings, or provided new insight, were identified until

saturation was achieved and included in the analysis. Participants did not provide feedback on the FGD findings.

### Ethical considerations

Ethical clearance was obtained from the Emory University Institutional Review Board and the Rwanda National Ethics Committee. Written informed consent was obtained from all study participants and all participants were informed about the nature and goals of the research. Each individual was compensated with 3,000 Rwandan Francs, an amount agreed upon by the local IRB.

## Results

### Quantitative strand

**Demographic and reproductive characteristics.** The case-control study included 165 postpartum women. PPIUD users (n = 74) were surveyed during PPIUD follow-up visits (n = 54) and infant vaccination (n = 20) services. PPIUD nonusers (n = 91) were surveyed during postpartum (n = 26) and infant vaccination services (n = 65) (S1 Fig).

There were no significant differences between groups in age, marital status, cohabitation, having health insurance, religion, education level, income, parity, or future desire for more children (Table 1). Over one in five (22%) women indicated that their most recent contraceptive method prior to pregnancy was a LARC, while 45% had not been using contraception. Receiving promotions at L&D (81% vs 46%, $P < .001$) was significantly associated with PPIUD uptake with non-PPIUD users more likely to report promotions at ANC (61% PPIUD vs 78% non-PPIUD, $P = .016$), infant vaccination (22% vs 47%, $P = .001$), and postpartum (19% vs 33%, $P = .042$) were. Across both groups, 62% of respondents had planned their recent pregnancy and 90% planned to breastfeed exclusively. Exclusive male partner control over FP decision making was associated with non-PPIUD use (relative to woman only control or joint decision making ($P = .015$). More users than nonusers reported that their fertility plans impacted their family planning decisions (72% vs 9%, $P < .001$). About 29% of nonusers who were not using contraception at the time of the survey planned to use an IUD within 3 months of the survey.

**Family planning and PPIUD knowledge.** There were notable family planning and contraceptives knowledge gaps among all respondents (Table 2). When asked about the recommended time for pregnancy after birth, 48% of the women correctly reported at least 2 years. When prompted to support their answer, 55% of respondents explained that birth spacing is important to ensure child health, to allow the mother's body to recuperate from the stress of pregnancy, and to have sufficient time for breastfeeding. When asked an open-ended question about what they had learned about the PPIUD, women reported that it was nonhormonal (87%), long-term (55%), highly effective (31%), reversible (17%), easy to remove whenever needed (9%), easily inserted after delivery (9%), and that you only had to pay once for the IUD compared to paying per provision when using injectables or pills (6%). While overall knowledge was low, users, compared to non-users, were more likely to spontaneously volunteer, without being prompted, that the IUD can be easily inserted immediately after delivery (14% vs 4%, $P = .037$), required no further action once inserted (14% vs 4%, $P = .037$), easy to remove when needed (16% vs 3%, $P = .004$), paying once for IUD compared to paying per provision when using injectables or pills (12% vs. 1%, $P = .003$).

When asked open-ended questions about the possible side-effects associated with PPIUD use, 39% reported cramping and backache, heavy periods after menses (25%), spotting between periods or heavy periods (18%), and other (30%). Among the women who chose "Other" for side-effects, an open response option was provided; 55% reported that the PPIUD

**Table 1. Sociodemographic, fertility and reproductive characteristics of respondents by PPIUD use (N = 165).**

| Variables | Total (N = 165) | PPIUD Users (n = 74) | PPIUD Nonusers (n = 91) | P value |
|---|---|---|---|---|
| **Age, mean (SD), y** | 28.9 (5.7) | 28.3 (6.3) | 29.4 (5.2) | .24 |
| **Relationship Status, n (%)** | | | | |
| Married | 127 (77.0) | 54 (73.0) | 73 (80.2) | .27 |
| Unmarried (Single, Divorced/Separated, Widow, Other) | 38 (23.0) | 20 (27.0) | 18 (19.8) | |
| **Living Situation, n (%)** | | | | |
| Cohabiting with partner | 144 (87.3) | 63 (85.1) | 81 (89.0) | .46 |
| Other (alone, with parents/family, roommates) | 21 (12.7) | 11 (14.9) | 10 (11.0) | |
| **Religion, n (%)** | | | | |
| Catholic | 48 (29.1) | 26 (35.1) | 22 (24.2) | .28 |
| Pentecostal | 78 (47.3) | 31 (41.9) | 47 (51.6) | |
| Other (Seventh Day Adventists, Jehovah's Witnesses, Anglican, Baptist, Muslim, Other, None) | 39 (23.6) | 17 (23.0) | 22 (24.2) | |
| **Education Level, n (%)** | | | | |
| No schooling | 26 (15.8) | 11 (14.9) | 15 (16.5) | .96 |
| Primary | 79 (47.9) | 36 (48.6) | 43 (47.3) | |
| Other (Secondary, College/University) | 60 (36.4) | 27 (36.5) | 33 (36.3) | |
| **Work or exchange services for money, n (%)** | | | | |
| Yes | 75 (45.5) | 36 (48.7) | 39 (42.9) | .46 |
| No | 90 (54.6) | 38 (51.4) | 52 (57.1) | |
| **Mutuelle (Government Health Insurance), n (%)** | | | | |
| Yes | 155 (93.9) | 67 (90.5) | 88 (96.7) | .11 |
| No[a] | 10 (6.1) | 7 (9.5) | 3 (3.3) | |
| **Parity, n (%)** | | | | |
| 0–1 | 47 (28.5) | 23 (31.1) | 24 (26.4) | .77 |
| 2–3 | 77 (46.5) | 34 (45.9) | 43 (47.3) | |
| 4 or more | 41 (24.8) | 17 (23.0) | 22 (26.4) | |
| **No. of living children, mean n (%)** | | | | |
| 0–1 | 50 (30.3) | 23 (31.1) | 27 (29.7) | .98 |
| 2–3 | 83 (50.3) | 37 (50.0) | 46 (50.5) | |
| 4 or more | 32 (19.4) | 14 (18.9) | 18 (19.8) | |
| **Most recent contraception method prior to pregnancy, n (%)** | | | | |
| LARC (Copper T-IUD, Implant) | 36 (21.8) | 20 (27.0) | 17 (18.7) | .32 |
| Other (Condoms, Pills, Injectables) | 55 (33.3) | 25 (33.8) | 29 (31.9) | |
| Never used contraception | 74 (44.8) | 29 (39.2) | 45 (49.5) | |
| **PPIUD Promotion Service Venues[b], n (%)** | | | | |
| Antenatal Care | 116 (70.3) | 45 (60.8) | 71 (78.0) | .016 |
| Infant Vaccination | 59 (35.8) | 16 (21.6) | 43 (47.3) | .001 |
| Labor & Delivery | 102 (61.8) | 60 (81.1) | 42 (46.2) | < .001 |
| Postpartum (before discharge) | 44 (26.7) | 14 (18.9) | 30 (33.0) | .042 |
| Community Health Worker | 7 (4.2) | 2 (2.7) | 5 (5.5) | .46 |
| **Was the most recent pregnancy planned? n (%)** | | | | |
| Yes | 102 (61.8) | 45 (60.8) | 57 (62.6) | .81 |
| No | 63 (38.2) | 29 (39.2) | 34 (37.4) | |
| **Breastfeeding Plans, n (%)** | | | | |
| Yes, exclusively | 149 (90.3) | 69 (93.2) | 80 (87.9) | .25 |
| Yes, non-exclusively | 16 (9.7) | 5 (6.8) | 11 (12.1) | |
| **Final Decision Maker regarding contraception, n (%)** | | | | |

*(Continued)*

**Table 1.** (Continued)

| Variables | Total (N = 165) | PPIUD Users (n = 74) | PPIUD Nonusers (n = 91) | P value |
|---|---|---|---|---|
| Me & My Partner and I | 139 (84.2) | 68 (91.9) | 71 (78.0) | .015 |
| My Partner | 26 (15.8) | 6 (8.1) | 20 (22.0) | |
| **Desire More Children, n (%)** | | | | |
| Yes | 91 (55.2) | 36 (48.6) | 55 (60.4) | .13 |
| No/undecided | 74 (44.9) | 38 (51.4) | 36 (39.6) | |
| **Did your fertility Plans Impact FP Decision?, n (%)** | | | | < .001 |
| Yes | 61 (37.0) | 53 (71.6) | 8 (8.8) | |
| No | 104 (63.0) | 21 (28.4) | 83 (91.2) | |

Abbreviations: CHW, community health worker; FP, family planning; IUD, intrauterine device; PPIUD, postpartum intrauterine device; PPLARC, postpartum long-acting reversible contraception; SD, standard deviation, USD, United States Dollar.

P value derived from two-tailed independent sample t-test for continuous variables and chi-square test for categorical variables (or Fisher's exact test for categorical variables with 20% of expected cell counts less than 5).

[a] Includes insurance that is not mutuelle.

[b] Select all that apply.

**Table 2. Knowledge about family planning and the PPIUD among respondents by PPIUD use.**

| Variables | Total (N = 165) | PPIUD Users (n = 74) | PPIUD Nonusers (n = 91) | P value |
|---|---|---|---|---|
| **Recommended Pregnancy spacing after last birth, n (%)** | | | | |
| No limit | 5 (3.0) | 4 (5.4) | 1 (1.1) | .18 |
| 2 years | 80 (48.5) | 33 (44.6) | 47 (51.6) | |
| I Don't Know or Don't remember | 52 (31.5) | 21 (28.4) | 31 (34.1) | |
| Other [ab] | 28 (17.0) | 16 (21.6) | 12 (13.2) | |
| **Reason for recommended pregnancy spacing after last birth (open-ended question)[a] (n = 112), n (%)** | | | | |
| To ensure healthy growth of child | 62 (55.4) | 21 (40.4) | 41 (68.3) | .003 |
| Other[d] | | | | |
| **PPIUD Knowledge after PPIUD counseling[c, g], n (%)** | | | | |
| Highly effective | 51 (30.9) | 28 (37.8) | 23 (25.3) | .082 |
| Long-term | 90 (54.5) | 38 (51.4) | 52 (57.1) | .46 |
| Reversible | 28 (17.0) | 16 (21.6) | 12 (13.2) | .15 |
| Can be easily inserted immediately after delivery | 14 (8.5) | 10 (13.5) | 4 (4.4) | .037 |
| Doesn't use hormones | 143 (86.7) | 66 (89.2) | 77 (84.6) | .39 |
| Once inserted no other action required from client to prevent pregnancy | 14 (8.5) | 10 (13.5) | 4 (4.4) | .037 |
| Easy to remove whenever needed | 15 (9.1) | 12 (16.2) | 3 (3.3) | .004 |
| If you use injectable or pills, you need to pay per provision while IUD you pay only once | 10 (6.1) | 9 (12.2) | 1 (1.1) | .003 |
| Other[e] | 29 (17.6) | 12 (16.2) | 17 (18.7) | .68 |
| **Possible PPIUD Side Effects[c], n (%)** | | | | |
| Cramping and backache for few days after insertion | 64 (38.8) | 32 (43.2) | 32 (35.2) | .29 |
| Spotting between periods or heavy periods | 29 (17.6) | 12 (16.2) | 17 (18.7) | .68 |
| Heavy periods after menses | 41 (24.8) | 17 (23.0) | 24 (26.4) | .62 |
| Other | 49 (29.7) | 23 (31.1) | 26 (28.6) | .73 |
| **Recommended PPIUD Insertion Time[c, h], n (%)** | | | | |
| Post placental (<10mins) | 143 (86.7) | 67 (90.5) | 76 (83.5) | .19 |

*(Continued)*

**Table 2.** (Continued)

| Variables | Total (N = 165) | PPIUD Users (n = 74) | PPIUD Nonusers (n = 91) | P value |
|---|---|---|---|---|
| 10 mins– 48 hours Postpartum | 50 (30.3) | 23 (31.1) | 27 (29.7) | .84 |
| 4–6 weeks | 108 (65.5) | 41 (55.4) | 67 (73.6) | .014 |
| Other[f] | 28 (17.0) | 12 (16.2) | 16 (17.6) | .82 |
| **Recommended PPIUD Insertion Time, n (%)** | | | | |
| Correct: citation of all three: Post placental (<10mins), 10mins– 48 hours postpartum, 4–6 weeks | 33 (20.0) | 15 (20.3) | 18 (19.8) | .94 |
| Incorrect: all other responses | 132 (80.0) | 59 (79.7) | 73 (80.2) | |
| **Expulsion Possible, n (%)** | | | | |
| Yes | 101 (61.2) | 49 (66.2) | 52 (57.1) | .23 |
| No | 64 (38.8) | 25 (33.8) | 39 (42.9) | |
| **How PPIUD prevents pregnancy** (n = 163), **n (%)** | | | | .91 |
| It stops the meeting of the spermatozoa and ovum | 99 (60.7) | 44 (60.3) | 55 (61.1) | |
| I Don't Know/Don't Remember/Other | 64 (39.3) | 29 (39.7) | 35 (38.9) | |
| **Heard about PPIUD prior to counseling, n (%)** | | | | |
| Yes | 41 (25) | 17 (23.0) | 24 (26.0) | .62 |
| No | 124 (75) | 57 (77.0) | 67 (67.0) | |

[a] Coded qualitative response.

[b] "Other" option includes 3, 4, 5, and 10 years.

[c] Select all that apply.

N's may not add to totals due to missingness of data for questions that participants were not required to answer.

[d] Other reasons for recommended pregnancy time after birth include: mother's body can prepare for next baby, improve overall health of family (economically and physically), birth spacing, enough time for breastfeeding.

[e] Other PPIUD knowledge include: IUD doesn't have many side effects, no frequent follow-up visits, may be removed at any time.

[f] Other responses for side effects include: None, I Don't Know/I don't remember.

[g] Non-prompted, spontaneous responses from women

[h] Respondents were read the various response options

had no side-effects and 25% did not know or remember. When asked to select from various response options which were provided about what time frame PPIUD can be inserted, respondents identified post-placental (87%), 10 mins–48 hours postpartum (30%), and 4–6 weeks (66%), however only 30 (20%) of the women correctly reported that PPIUDs can be inserted at all 3 times. Women who reported that the PPIUD can be inserted only between 4–6 weeks were less likely to be users (55% vs. 74%, $P$ = .014). Only 61% indicated that expulsion was possible. When asked to explain how the method prevents pregnancy, 61% reported that it stops the spermatozoa and ovum from meeting. Regarding women's general awareness about the PPIUD, 25% had heard about the method previously from informal social networks such as peers, neighbors, and other users.

**Reasons for acceptance or rejection of the PPIUD.** Women were also asked about factors that impacted their decisions to either accept or reject the PPIUD. Among users, the top factors indicated were that the method is nonhormonal (74%), long-term (58%), and highly effective (38%) (Table 3). Almost all users (99%) reported that they found the promotions useful in their decisions, and 88% stated that they would not have gotten the method without them. Partner's rejection of PPIUD (28%), absence of partner during decision time (19%), religion (14%), and influence from other women (13%) were cited as reasons for PPIUD non-uptake. Other reasons reported by nonusers were delivering at a different facility, myths about

**Table 3. Reasons for acceptance and rejection of PPIUD.**

| Reasons for PPIUD Acceptance [a] | n (%) |
|---|---|
| Doesn't use hormones | 55 (74.3) |
| Long-term | 43 (58.1) |
| Highly effective | 27 (36.5) |
| Other [b] | 13 (17.6) |
| Reversible | 11 (14.9) |
| Once inserted no other action required from client to prevent pregnancy | 11 (14.9) |
| Can be easily inserted immediately after delivery | 9 (12.2) |
| Easy to remove whenever needed | 9 (12.2) |
| If you use injectable or pills, you need to pay per provision while IUD you pay only once | 9 (12.2) |
| Used to be happy on it/Happy testimony | 9 (12.2) |
| **PPIUD promotions useful in decision making?** | |
| Yes | 73 (98.6) |
| No | 1 (1.4) |
| **Would you have gotten the PPIUD anyway without promotions?** | |
| No | 65 (87.8) |
| Yes | 9 (12.2) |
| **Reasons for PPIUD Rejection** [a] | |
| Other[c] | 29 (31.9) |
| My partner refused to use PPIUD or doesn't like it | 25 (27.5) |
| My partner was not present | 17 (18.7) |
| Religious reasons | 13 (14.3) |
| Influence from women in the same room | 12 (13.2) |
| Unhappy peer testimony | 4 (4.4) |
| Side-effects | 1 (1.1) |
| **Why did you choose your current method?** | |
| Preference for current method and its benefits | 51 (56.0) |
| None/no reason | 18 (19.8) |
| Partner's choice | 6 (6.6) |
| Insufficient decision-making time | 5 (5.5) |

[a] Select all that apply so percentages are more than 100%.

[b] "Other" responses include: Little/no side effects compared to other methods, secrecy of where IUD is placed.

[c] "Other" responses include: delivering at different facility, rumors and myths (cancer, tumor, genital infections, discomfort during sexual intercourse, pregnancy while on IUD).

the IUD being cancer-causing, genital infections, discomfort during sexual intercourse and pregnancy while on method were cited as reasons for PPIUD non-uptake.

**Multivariate regression.** Variables significant at $P<0.01$ were included in the final multivariate logistic model. Women who reported that PPIUD can be inserted 4–6 weeks after delivery were less likely to uptake the method than those who did not (adjusted odds ratio [aOR], 0.17; 95% CI, 0.06–0.44; $P < .001$) (Table 4). Compared to women whose partner alone made the final PPFP decision, women who made independent and joint decisions with their partners were more likely to uptake the PPIUD (aOR, 4.04; 95% CI, 1.12–14.6; $P = .033$). Finally, the odds of accepting the PPIUD were higher among women who reported taking their fertility intentions into account when making their PPFP decisions versus those who did not (aOR, 48.5; 95% CI, 16.4–143.4; $P < .001$).

**Table 4. Multivariate analysis of factors associated with PPIUD use.**

| Variables | Bivariate analysis | | Logistic regression analysis | | | |
|---|---|---|---|---|---|---|
| | cOR (95% CI) | P value | Full model aOR (95% CI) | P value | Reduced model aOR (95% CI) | P value |
| **Recommended Pregnancy Time after birth** | | | | | *Not included* | |
| Other | 1 [Reference] | | 1 [Reference] | | | |
| No limit | 3.00 (0.30–30.4) | .17 | 4.09 (0.19–87.7) | .30 | | |
| At least 2 years | 0.53 (0.22–1.26) | .092 | 0.73 (0.20–2.64) | .24 | | |
| I Don't Know or Don't remember | 0.51 (0.20–1.29) | .091 | 0.91 (0.23–3.55) | .51 | | |
| **PPIUD Knowledge (yes versus no)** | | | | | *Not included* | |
| Highly effective | 1.80 (0.92–3.51) | .084 | 2.01 (0.66–6.1) | .22 | | |
| Can be easily inserted immediately after delivery | 3.40 (1.02–11.3) | .046 | 1.39 (0.21–9.09) | .73 | | |
| Once inserted no other action required from client to prevent pregnancy | 3.40 (1.02–11.3) | .046 | 2.20 (0.35–13.9) | .40 | | |
| Easy to remove whenever needed | 5.68 (1.54–21.0) | .009 | 2.22 (0.33–14.8) | .41 | | |
| If you use injectable or pills, you need to pay per provision while IUD you pay only once | 12.5 (1.54–100.8) | .018 | 0.93 (0.05–16.8) | .96 | | |
| **Knowledge of PPIUD insertion time option (yes versus no)** | | | | | | |
| 4–6 weeks | 0.45 (0.23–0.86) | .015 | 0.13 (0.05–0.39) | < .001 | 0.17 (0.06–0.44) | < .001 |
| **Final Decision Maker regarding PP FP** | | | | | | |
| My partner | 1 [Reference] | .019 | 1 [Reference] | .095 | 1 [Reference] | .033 |
| Me & My Partner and I | 3.19 (1.21–8.43) | | 3.21 (0.82–12.6) | | 4.04 (1.12–14.6) | |
| **Fertility Plans Impact FP Decision** | | | | | | |
| No | 1 [Reference] | < .001 | 1 [Reference] | < .001 | 1 [Reference] | < .001 |
| Yes | 26.2 (10.8–63.4) | | 44.4 (14.2–138.9) | | 48.5 (16.4–143.4) | |

Abbreviations: aOR, adjusted odds ratio; CI, confidence interval; cOR, crude odds ratio; FP, family planning; IUD: intrauterine device; OR, odds ratio; PP, post-partum; PPIUD, postpartum intrauterine device.

P value <0.1

## Qualitative strand

Forty-one promoters participated in focus groups (17 CHWs and 24 nurses, midwives and physicians). The data highlighted the following themes that complemented survey findings: low PPIUD knowledge, the importance of male partner involvement in IUD decision-making, rumors and myths, concerns and side effects, inconsistent messages from providers, and perception of intervention as factors that impacted acceptance or rejection of PPIUD.

**Low PPIUD knowledge.** Providers reported that clients had low IUD-specific knowledge, including benefits of postpartum insertion, the insertion and removal process, how it prevents pregnancy, and why it is highly efficacious relative to other contraceptives. Providers shared that clients are generally suspicious about the IUD as it is new to many Rwandan women, and its placement through the cervix makes them uncomfortable. It is easier for women to trust contraceptive methods like pills and implants because they are visible and palpable unlike an IUD.

Providers explained that for some women, especially young first-time mothers, unfamiliarity with the female anatomy and medical terms like "placenta" and "cervix" cause confusion. Some women mistake the cervix for the vagina and do not understand how a metal placed in the vagina will not expulse or cause harm. To address this fear, a hospital provider stated [we use] *didactic materials, an actual IUD and the pictures within the flip chart [to] facilitate the*

*promotion to show them what we are explaining.*" They also explain that the post-placental insertion time is ideal because the cervix is large and open, making the device easier to place, without causing pain or discomfort. Providers reported that they emphasize the nonhormonal, long-acting and effective nature of the PPIUD.

**Male partner involvement in IUD decision-making.** Providers discussed the important role men can play in a woman's decision to use an IUD.

Some women hear about the PPIUD during ANC visits and inform their partners about it, who later accompany them to the follow-up visits to learn more about it. According to the providers, couples counseling during ANC is the most effective time to deliver the intervention because the couples have sufficient time to ask questions and receive information that will lead to informed decisions. A nurse stated, *"During the couple counseling the promotion is easier compared to when a woman is alone, because men are more convinced on family planning than women."* Once they are counseled and understand the benefits like the economic and financial impact, they are more "*receptive and supportive.*"

**Rumors and myths.** Providers explained that some women may decide to uptake the PPIUD during ANC visits. However, rumors and myths (i.e., the IUD is harmful, moves to other parts of the body, aborts pregnancies, causes infertility) circulated by peers and neighbors during community gatherings like community workdays may dissuade women from adopting the method. For example, a HC provider stated: "*Rumors and myths that women have include: IUDs can disappear, cause discomfort during sexual intercourse, or cause infection and cancer. . .so for this [reason] it is not easy to convince them [to use an IUD]."* These rumors may perpetuate fear and cause distrust between women and their HC providers. To dispel these rumors, providers explained that increasing the use of satisfied IUD user testimonies and allowing medical staff to go into communities for PPFP promotion would be beneficial.

**Concerns and side-effects.** Providers felt PPIUD fears and concerns are prompted by peers, neighbors and unhappy users. A HC provider described that segments of the flipchart that address side-effects prompted clients to ask the most questions. He stated, *"the part of side-effects raises more questions . . .they [the clients] ask you if IUD [has] side-effects like weight gain, infection, backache. . . mostly [these questions are asked by] those who have had side-effects on other contraceptive methods."* Some clients expressed that they would experience a reduction of vaginal secretions during sexual intercourse because of the IUD. Providers explained that for women who undergo difficult deliveries that lead to tears or episiotomy, fear about additional pain during the IUD insertion is a concern. The reversibility of the PPIUD also raised concerns for some women who plan to have more children. According to a CHW, *"some clients think that only the person who inserts the IUD can take it out."* To address these concerns, providers explained to the women that any trained staff can remove their IUD at their request. Moreover, providers expressed that some clients were concerned about inconsistent messages (e.g., that pills and injectables are nonhormonal or that the IUD is an abortifacient).

**Perception of intervention.** Providers spoke to the strength of the PPIUD service delivery and trainings they received which facilitates their ability to counsel clients effectively. They also underscored the importance of the continuity and repetition of messages during units of care like ANC, L&D, infant vaccination and follow-up. Participants also highlighted the fact that the PPIUD flipchart is a helpful resource to refer to and confirm the information they deliver. A provider suggested having an educational video playing in the waiting rooms to aid in PPIUD promotions, especially when providers are busy with other activities. Another recommended "*distributing individual brochures may help in the promotion as a woman can continue reading the information once home, and if she hears rumors, she may again refer to the brochure.*"

## Discussion

This study employed a mixed-methods approach which found consistency between qualitative and quantitative data to understand factors associated with PPIUD uptake after PPFP education. Woman alone or joint FP decision making was associated with uptake relative to exclusive male partner control. Additionally, not being able to spontaneously report that the IUD can be inserted immediately post-partum or within 48 hours of delivery was associated with non-uptake of the method. PPIUD users reported that the provision of PPFP counseling at the PSF-affiliated facilities was critical in their decision to uptake an IUD. This demonstrates the strength of the intervention and highlights the need for more PPIUD competent providers.

According to WHO, PPFP counseling should ideally begin during ANC, however initial counseling during early labor and immediate postpartum are also acceptable [20]. In this study, 61% and 81% of women reported being counseled in ANC and L&D, respectively. Providers explained that once promotions have occurred during ANC, subsequent promotions are easier because they are able to address residual questions and concerns and emphasize the method's advantages. Other literature similarly reports that repeated counseling during the ANC and postpartum periods can positively impact contraceptive use [21].

A WHO report showed that a common reason for nonuse of contraception is lack of awareness [20]. Findings from the study show that the majority (77%) of the study population were not aware of the PPIUD prior to PPFP counseling. Contraceptive provision in many sub-Saharan African countries has focused predominantly on short-terms methods such as condoms, injectables, and pills [22–26]. Preference for other contraceptive methods and their benefits was also a common reason for nonuse.

Overall, PPIUD knowledge gaps persisted after PPFP counseling. Only 20% of women could correctly identify all the possible PPIUD insertion times. After receipt of PPFP counseling, <9% of respondents spontaneously (without prompting) reported that the IUD can be easily inserted immediately after delivery, requires fewer visits, and is removable whenever needed. A Tanzanian study on women's perspectives on, and experiences of using the PPIUD suggested that women's limited knowledge of PPIUD advantages may have stemmed from incomplete contraceptive counseling [27]. These findings highlight domains for further improvement of PPFP counseling. When faced with a new method that is different in several ways from more familiar methods, it is also challenging to retain multiple pieces of information after only one presentation.

Most women were motivated to use the PPIUD due to its lack of hormone-related side effects, effectiveness, ability to breastfeed, reduced follow-up visits and duration of protection against pregnancy; these findings are consistent with other studies [24, 28]. According to providers, women expressed misconceptions and concerns related to PPIUD use, including the risk of cancer, negative impact on sexual experiences, possible infections, and pain. Fear of side effects, and longstanding myths and misconceptions such as infection and infertility about the IUD have been associated with declining the method in other studies [24, 25, 29–31]. Previous studies also show that IUD misinformation has been spread by local informal social networks, unhappy peer IUD users, and through religious authorities [25, 30, 32–34]. Providers can be trained to address these concerns during counseling.

Partner involvement was a salient factor in women's decisions; 46% of PPIUD nonusers indicated that partner's absence and refusal of the method led to their rejection of the PPIUD. This finding was echoed by providers who expressed that partner presence during counseling impacts women's FP decisions. Numerous studies have also found partner's involvement to be very fundamental in FP decisions in many sub-Saharan African countries [24, 25, 29, 31, 34, 35]. For example, in a study of 1,914 pregnant women in Ghana, partner attitudes (specifically

positive attitudes towards LARC) were a key component in PPFP decision-making for 80% of couples [35]. It may be highly beneficial to provide the PPIUD educational intervention during the first ANC visit, which >80% of Rwandan male partners attend [36].

Some limitations warrant consideration. Though this is one of few exploratory studies to incorporate both client and provider perspectives related to PPIUD use, the small sample size for the case-control study limits our ability to rule out type II error. The sample is more generalizable to urban women who have access to healthcare. It is possible that the nonusers surveyed prior to 6 weeks had intentions to, or did, take up the PPIUD within the 6-week study time frame. Additionally, the number of women approached to participate who declined was not recorded thus a response rate could not be obtained. Our focus groups size was larger than is customary to ensure proper elicitation of responses from all participants, though our facilitators had many years of experience conducting similar focus groups. Lastly, social desirability bias when collecting self-reported information is possible.

## Conclusion

Taking the service delivery perspective into consideration when developing family planning programs may be important as providers can reveal nuanced information that may not be provided by clients. The present study highlighted knowledge of and concerns about the PPIUD as well as how male partner involvement in family planning decisions may influence PPIUD uptake. Widespread campaigns about the IUD's advantages and safety, and proactive counseling to address couples' specific fears may increase awareness and uptake of the method. These efforts could aim to improve knowledge and overcome misconceptions related to all available contraceptives with the ultimate goal of improved client-centered counseling and joint decision-making. In the Rwandan context, community promotions can occur during community events led by CHWs. Our findings may inform how to iteratively refine our already successful PPFP intervention which can be scaled-up to meet the contraceptive needs of postpartum women in Rwanda.

## Supporting information

**S1 Fig. Flowchart for case-control study participant recruitment.**
(DOCX)

**S1 Questionnaire. Inclusivity in global research questionnaire.**
(DOCX)

## Author Contributions

**Conceptualization:** Mariama S. Tounkara, Susan Allen, Lisa B. Haddad, Vanessa Da Costa, Kristin M. Wall.

**Data curation:** Mariama S. Tounkara, Rachel Parker.

**Formal analysis:** Mariama S. Tounkara.

**Funding acquisition:** Susan Allen, Kristin M. Wall.

**Investigation:** Mariama S. Tounkara, Rosine Ingabire, Etienne Karita, Julien Nyombayire, Vanessa Da Costa, Amelia Mazzei, Kristin M. Wall.

**Methodology:** Mariama S. Tounkara, Rosine Ingabire, Dawn L. Comeau, Etienne Karita, Susan Allen, Julien Nyombayire, Rachel Parker, Lisa B. Haddad, Vanessa Da Costa, Amelia Mazzei, Kristin M. Wall.

**Project administration:** Mariama S. Tounkara, Rosine Ingabire, Etienne Karita, Julien Nyombayire, Amanda Tichacek, Amelia Mazzei, Jeannine Mukamuyango, Kristin M. Wall.

**Supervision:** Rosine Ingabire, Dawn L. Comeau, Etienne Karita, Susan Allen, Julien Nyombayire, Rachel Parker, Amanda Tichacek, Amelia Mazzei, Jeannine Mukamuyango, Kristin M. Wall.

**Writing – original draft:** Mariama S. Tounkara.

**Writing – review & editing:** Rosine Ingabire, Dawn L. Comeau, Etienne Karita, Susan Allen, Julien Nyombayire, Rachel Parker, Lisa B. Haddad, Vanessa Da Costa, Amanda Tichacek, Amelia Mazzei, Jeannine Mukamuyango, Kristin M. Wall.

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
