## [Decision Letter · Decision Letter 0]

8 Aug 2022

PONE-D-22-16684A mixed-methods study of factors influencing postpartum intrauterine device uptake after family planning counseling among women in Kigali, RwandaPLOS ONE

Dear Dr. Wall,

Thank you for submitting your manuscript to PLOS ONE. After careful consideration, we feel that it has merit but does not fully meet PLOS ONE’s publication criteria as it currently stands. Therefore, we invite you to submit a revised version of the manuscript that addresses the points raised during the review process.

We look forward to receiving your revised manuscript.

Kind regards,

Kehinde Sharafadeen Okunade

Academic Editor

PLOS ONE

Journal Requirements:

“Financial support for this work was provided by the Bill & Melinda Gates Foundation, Emory University Research Council Grants, Emory Global Field Experience Program, National Institutes of Health and Emory AITRP Fogarty.”

“This work was supported by the Bill & Melinda Gates Foundation [OPP1160661]. Additional support came from the Emory University Research Council Grant [URCGA16872456], Emory Global Field Experience Award, the Emory Center for AIDS Research [P30 AI050409], the National Institutes of Health [NIAID R01 AI51231; NIAID R01 AI64060; NIAID R37 AI51231], and Emory AITRP Fogarty [5D43TW001042]. The funders had no role in study design, data collection and analysis, decision to publish, or preparation of the manuscript.”

7. Please include your tables as part of your main manuscript and remove the individual files. Please note that supplementary tables (should remain/ be uploaded) as separate "supporting information" files

Reviewers' comments:

Reviewer's Responses to Questions

**Comments to the Author**

1. Is the manuscript technically sound, and do the data support the conclusions?

Reviewer #1: Partly

Reviewer #2: Partly

2. Has the statistical analysis been performed appropriately and rigorously? 

Reviewer #1: Yes

Reviewer #2: No

3. Have the authors made all data underlying the findings in their manuscript fully available?

Reviewer #1: No

Reviewer #2: Yes

4. Is the manuscript presented in an intelligible fashion and written in standard English?

Reviewer #1: Yes

Reviewer #2: No

5. Review Comments to the Author

Reviewer #1: General Comments:

Overall, the manuscript is well written. However, the meaning certain abbreviations should be clearly defined such L & D.

Specific Comments

Study design:

Although the choice of case-control study design was appropriate for the users and non-users of postpartum IUD, the sampling techniques and the ratio of users and non-users selected for study was not stated.

Results Section:

The tables referenced in the result section were not available for review in the submitted manuscript.

Reviewer #2: A mixed-methods study of factors influencing postpartum intrauterine device uptake after family planning counseling among women in Kigali, Rwanda

This study which aimed to ascertain factors that influenced postpartum intrauterine device uptake after family planning counseling among women in Kigali, Rwanda is quite relevant towards improving PPIUD uptake in developing countries which is vital for reduction of preventable maternal deaths.

Overall, the article has been fairly well written but there are sections that needs to be revised to improve the quality of the manuscript.

The quick point subsection should

Abstract

Abstract has not been written in accordance with journal specification for the subsections under the abstract. Please change discussion to conclusion

The result section stated that factors associated with PPIIUD (P-value <0.1) was what was used to make inferences. This should be reviewed and aligned with what has been written in the main text in which P-value P value <0.1 was considered for adjusted logistic regression analysis and not at the bivariate level

Conclusion is misleading due to the value of level of significance for which the null hypothesis will be rejected or accepted. (High chance of making type II error)

METHODS

Page 6, Lines 48-51: Paraphrase to:

Sites were selected because they were high volume facilities where CHWs and trained PPFP providers rendered PPFP services that included counseling in ANC, labor and delivery, postpartum (before discharge) and at 6-week infant vaccination visits

Page 9, lines 94-96:

Comment: The number of participants in this FGD seems too many to allow for a controlled and enabling focused discussion. Ideally it should not be more than 10 participants per FGD group. This is because large groups are difficult to control and they limit each person’s opportunity to make their observations or give their perspectives. This should be mentioned as a limitation to the study. How did the authors overcome this challenge?

Page 11, line 113

Comment: The reason for the compensation should be justified with a sentence.

Page 11, line 114-115

Comment: This was a case-control study; Therefore, the reports should be comparing the cases and the controls. Was the over 22% amongst women using PPPIUD currently or the non-users?

Page 13, lines 174-175

Comment: Rather than lump them as others which is not entirely true based on the responses. For example, not knowing or remembering a side effect doesn’t make it "others" and saying that there are no side effects doesn’t make it a type of side effect.

Page 14, line 198: Variables significant at P<0.01

Comment: This different from P<0.1 earlier presented in the methodology section. Please reconcile the two statements

Qualitative data:

This has not been reported appropriately: Please use the COREQ checklist and attach same as an appendix.

One important group which will answer the research question in the qualitative component of the study was conspicuously missing. This are the women who are eligible for PPIUD insertion. The omission of this group was a missed opportunity for a rich qualitative data from the perspective of the patients. This can be included in the limitation section

Tables

This was not attached in the manuscript

6. PLOS authors have the option to publish the peer review history of their article (what does this mean?). If published, this will include your full peer review and any attached files.

Reviewer #1: No

Reviewer #2: **Yes: **Dr Godwin Akaba,MBBS,MSc,MPH,FWACS

---

## [Author Response · Author response to Decision Letter 0]

20 Sep 2022

Reviewer #1: General Comments:

Overall, the manuscript is well written. However, the meaning certain abbreviations should be clearly defined such L & D.

We have reviewed all abbreviations to make sure they are defined. 

Specific Comments

Study design:

Although the choice of case-control study design was appropriate for the users and non-users of postpartum IUD, the sampling techniques and the ratio of users and non-users selected for study was not stated.

We now state that this study relied on a convenience sample with a target ratio of 1:1. 

Results Section:

The tables referenced in the result section were not available for review in the submitted manuscript.

Apologies, I’m not sure why the tables were not viewable to the reviewers.

Reviewer #2: A mixed-methods study of factors influencing postpartum intrauterine device uptake after family planning counseling among women in Kigali, Rwanda

This study which aimed to ascertain factors that influenced postpartum intrauterine device uptake after family planning counseling among women in Kigali, Rwanda is quite relevant towards improving PPIUD uptake in developing countries which is vital for reduction of preventable maternal deaths.

We appreciate the reviewers positive and constructive comments. 

Overall, the article has been fairly well written but there are sections that needs to be revised to improve the quality of the manuscript.

The quick point subsection should

Abstract

Abstract has not been written in accordance with journal specification for the subsections under the abstract. Please change discussion to conclusion

This edit has been made. 

The result section stated that factors associated with PPIIUD (P-value <0.1) was what was used to make inferences. This should be reviewed and aligned with what has been written in the main text in which P-value P value <0.1 was considered for adjusted logistic regression analysis and not at the bivariate level.

We have ensured that the use of p<0.01 as the cut-off for statistical significance has been made consistently throughout the manuscript. 

Conclusion is misleading due to the value of level of significance for which the null hypothesis will be rejected or accepted. (High chance of making type II error)

We now temper our findings in light of the potential for type II error. 

METHODS

Page 6, Lines 48-51: Paraphrase to:

Sites were selected because they were high volume facilities where CHWs and trained PPFP providers rendered PPFP services that included counseling in ANC, labor and delivery, postpartum (before discharge) and at 6-week infant vaccination visits

This section has been paraphrased.

Page 9, lines 94-96:

Comment: The number of participants in this FGD seems too many to allow for a controlled and enabling focused discussion. Ideally it should not be more than 10 participants per FGD group. This is because large groups are difficult to control and they limit each person’s opportunity to make their observations or give their perspectives. This should be mentioned as a limitation to the study. How did the authors overcome this challenge?

This has been noted in the limitations section (though we also note that our FGS facilitators have been conducting FGDs for many years and are experienced at managing these discussions, we agree that these FGDs were larger than is customary). 

Page 11, line 113

Comment: The reason for the compensation should be justified with a sentence.

We now note that the compensation level was that approved/set by the local IRB to compensate participants for their time without being coercive. 

Page 11, line 114-115

Comment: This was a case-control study; Therefore, the reports should be comparing the cases and the controls. Was the over 22% amongst women using PPPIUD currently or the non-users?

The 22% is among the entire study population, and these data are described in table 1 stratified by PPIUD users and non-users. Apologies that the tables were not viewable by the reviewers.

Page 13, lines 174-175

Comment: Rather than lump them as others which is not entirely true based on the responses. For example, not knowing or remembering a side effect doesn’t make it "others" and saying that there are no side effects doesn’t make it a type of side effect.

We now clarify that ‘Other’ was an actual response option (not created post hoc), and choosing this option led to an open response option from which the responses were grouped as described in the text. 

Page 14, line 198: Variables significant at P<0.01

Comment: This different from P<0.1 earlier presented in the methodology section. Please reconcile the two statements

This inconsistency in definition of the p-value cutoff has been corrected.

Qualitative data:

This has not been reported appropriately: Please use the COREQ checklist and attach same as an appendix.

This was a helpful suggestion – we have modified the manuscript to adhere to this checklist, which has been attached in the resubmission. 

One important group which will answer the research question in the qualitative component of the study was conspicuously missing. This are the women who are eligible for PPIUD insertion. The omission of this group was a missed opportunity for a rich qualitative data from the perspective of the patients. This can be included in the limitation section.

We do not entirely understand this comment. All women studied were eligible for PPIUD insertion (there are very few contraindications to receiving the copper IUD at some point in the postpartum period). If more clarity on the comment can be provided so that the authors can understand the issue, we are happy to mention the issue in the limitations section.

Tables

This was not attached in the manuscript

Apologies, we are not sure why the tables were not viewable by the reviewers.

---

## [Editor Report · Decision Letter 1]

2 Oct 2022

A mixed-methods study of factors influencing postpartum intrauterine device uptake after family planning counseling among women in Kigali, Rwanda

PONE-D-22-16684R1

Dear Dr. Wall,

We’re pleased to inform you that your manuscript has been judged scientifically suitable for publication and will be formally accepted for publication once it meets all outstanding technical requirements.

Kind regards,

Kehinde Sharafadeen Okunade

Academic Editor

PLOS ONE
---

## [Editor Report · Acceptance letter]

12 Oct 2022

PONE-D-22-16684R1 

A mixed-methods study of factors influencing postpartum intrauterine device uptake after family planning counseling among women in Kigali, Rwanda 

Dear Dr. Wall:

I'm pleased to inform you that your manuscript has been deemed suitable for publication in PLOS ONE. Congratulations! Your manuscript is now with our production department. 

Kind regards, 

on behalf of

Dr. Kehinde Sharafadeen Okunade 

Academic Editor

PLOS ONE